# Retrospective Analysis for Dose Reduction to Organs at Risk with New Personalized Breast Holder (PERSBRA) in Left Breast IMRT

**DOI:** 10.3390/jpm12091368

**Published:** 2022-08-25

**Authors:** Chiu-Ping Chen, Tung-Ho Chen, Jeng-Fong Chiou, Yi-Ju Chen, Chia-Chun Kuo, Kuo-Hsiung Tseng, Meng-Yun Chung, Chun-You Chen, Jeng-You Wu, Long-Sheng Lu, Shih-Ming Hsu

**Affiliations:** 1Department of Radiation Oncology, Wan Fang Hospital, Taipei Medical University, Taipei 116, Taiwan; 2Department of Biomedical Imaging and Radiological Sciences, National Yang Ming Chiao Tung University, Taipei 112, Taiwan; 3Medical Physics and Radiation Measurements Laboratory, National Yang Ming Chiao Tung University, Taipei 112, Taiwan; 4Department of Radiation Oncology, Taipei Medical University Hospital, Taipei Medical University, Taipei 110, Taiwan; 5Department of Radiology, School of Medicine, College of Medicine, Taipei Medical University, Taipei 110, Taiwan; 6School of Health Care Administration, College of Management, Taipei Medical University, Taipei 110, Taiwan; 7Program for Cancer Molecular Biology and Drug Discovery, College of Medical Science and Technology, Taipei Medical University and Academia Sinica, Taipei 110, Taiwan; 8Department of Electrical Engineering, National Taipei University of Technology, Taipei 106, Taiwan; 9Graduate Institute of Biomedical Informatics, College of Medical Science and Technology, Taipei Medical University, Taipei 110, Taiwan; 10Graduate Institute of Data Science, Taipei Medical University, Taipei 110, Taiwan; 11Graduate Institute of Biomedical Materials and Tissue Engineering, College of Biomedical Engineering, Taipei Medical University, Taipei 110, Taiwan; 12TMU Research Center of Cancer Translational Medicine, Taipei Medical University, Taipei 110, Taiwan; 13International Ph.D. Program in Biomedical Engineering, College of Biomedical Engineering, Taipei Medical University, Taipei 110, Taiwan; 14International Ph.D. Program for Cell Therapy and Regenerative Medicine, College of Medicine, Taipei Medical University, Taipei 110, Taiwan

**Keywords:** breast cancer, personalized breast holder, organs at risk, intensity-modulated radiation therapy

## Abstract

This study evaluated dose differences in normal organs at risk, such as the lungs, heart, left anterior descending artery (LAD), right coronary artery, left ventricle, and right breast under personalized breast holder (PERSBRA), when using intensity-modulated radiation therapy (IMRT). This study evaluated the radiation protection offered by PERSBRA in left breast cancer radiation therapy. Here, we retrospectively collected data from 24 patients with left breast cancer who underwent breast-conserving surgery as well as IMRT radiotherapy. We compared the dose differences in target coverage and organs at risk with and without PERSBRA. For target coverage, tumor prescribed dose 95% coverage, conformity index, and homogeneity index were evaluated. For organs at risk, we compared the mean heart dose, mean left ventricle dose, LAD maximum and mean dose, mean left lung receiving 20 Gy, 10 Gy, and 5 Gy of left lung volume, maximum and mean coronary artery of the right, maximum of right breast, and mean dose. Good target coverage was achieved with and without PERSBRA. When PERSBRA was used with IMRT, the mean dose of the heart decreased by 42%, the maximum dose of LAD decreased by 26.4%, and the mean dose of LAD decreased by 47.0%. The mean dose of the left ventricle decreased by 54.1%, the volume (V_20_) of the left lung that received 20 Gy decreased by 22.8%, the volume (V_10_) of the left lung that received 10 Gy decreased by 19.8%, the volume (V_5_) of the left lung that received 5 Gy decreased by 15.7%, and the mean dose of the left lung decreased by 23.3%. Using PERSBRA with IMRT greatly decreases the dose to organs at risk (left lung, heart, left ventricle, and LAD). This study found that PERSBRA with IMRT can achieve results similar to deep inspiration breath-hold radiotherapy (DIBH) in terms of reducing the heart radiation dose and the risk of developing heart disease in patients with left breast cancer who cannot undergo DIBH.

## 1. Introduction

Breast cancer is one of the common cancers in women. Sung et al. stated that breast cancer accounts for 11.7% of all cancers in women [1]. With recent developments in diagnostic and cancer screening techniques, breast cancer can be diagnosed early. Breast-conserving surgery combined with postoperative radiation therapy is the recommended treatment for patients diagnosed in the early stages. Whole-breast radiation therapy after breast-conserving surgery can decrease local recurrence and mortality rates [2].

Whole-breast radiotherapy poses a great challenge for patients with left breast cancer due to the shape (concave) and position of the left breast, since it is located near the heart. In addition to tumor coverage, radiotherapy must be planned to protect organs at risk (lungs and heart) from receiving excessive radiation therapy doses as this may lead to radiation therapy-induced cardiovascular complications, such as ischemic heart disease, congestive heart failure, myocarditis, arrhythmia, and myocardial infarction [3]. In conventional tangential field treatment, a high dose will be irradiated in the position of the anterior heart, including the left anterior descending (LAD) coronary artery, which increases perfusion defects [4]. Darby (2013) revealed that every increase by 1 Gy in the mean heart dose increases the risk of developing coronary artery disease by 7.4% [5], while Taylor’s study in 2017 revealed that an increase in the mean heart dose by 1 Gy increases the probability of developing lung cancer by 11% [6]. Pneumonitis and lung fibrosis are potential complications that occur several months after radiotherapy [7]. Minimizing the dose in organs at risk (heart, lungs) decreases radiotherapy-induced complications and increases patient quality of life.

Deep inspiration breath-hold radiotherapy (DIBH) [8,9] or prone position [10,11] is often used in clinical practice to decrease the dose to the heart and lungs as the distance from the heart to the irradiation field increases. However, the prone position is associated with setup errors, leading to lower efficiency and stability [12]. Although DIBH is the better method to reduce irradiation to the heart or lungs [13], the criteria for DIBH stipulate that the patient should be able to hold their breath for at least 20 s [14]. Taken together, this raises the question of how to ensure that organs at risk are far from the irradiation field in a prone position while maintaining comfort and reproducibility in the supine position.

According to a study by Chen et al., the PERSBRA material needs more elasticity and biocompatibility; they used a thermoplastic elastomer (TPE) to fabricate a PERSBRA using a 3D printer. Patients wore the PERSBRA and a hybrid treatment regimen (80% conventional tangential field and 20% intensity-modulated radiation therapy (IMRT)) was used. They found that heart and lung doses were reduced by >20% with the PERSBRA compared to without [15]. Many studies have reported that IMRT has better tumor coverage than conventional tangential fields and decreases the dose to organs at risk [16,17,18]. In terms of evaluating the dose of radiotherapy in the left breast, most studies have compared the mean dose of the heart or the mean dose of the LAD [13,19,20,21] and few studies have examined doses to the substructures of the heart, left ventricle (LV), and right coronary artery, under different techniques.

The objective of this study was to evaluate the effects of a homemade PERSBRA with IMRT on breast cancer radiation therapy in terms of doses to organs at risk, including the lungs, heart, LAD artery, right coronary artery, LV, and right breast. The results of this retrospective study are used to evaluate the feasibility of PERSBRA with IMRT in whole left breast radiation therapy.

## 2. Materials and Methods

### 2.1. PERSBRA Design

This study used a personalized breast holder (PERSBRA) that was manufactured with a thermoplastic elastomer using a 3D printer. The patient was first placed in a prone position and we used a 3D scanner to acquire the image of the shape of the breast. This breast contour file was uploaded to a dedicated image processing system and a 3D printer was used to prepare a customized PERSBRA. According to the study by Chen et al., the customized PERSBRA fabrication took approximately 18–40 h using the 3D printer [22]. The patient was placed in a prone position while wearing the customized PERSBRA holder, undergoing daily treatment in the supine position (Figure 1). The patient was then placed in a supine position for each treatment, which showed better treatment reproducibility. PERSBRA created a similar treatment posture to the prone position to keep the heart and lung away from the treatment field.

### 2.2. Patient Selection

This retrospective study collected data from 24 women with left breast cancer from 2017 to 2021. All 24 patients underwent breast-conserving surgery and total breast radiotherapy postoperatively. All patients in the clinical trial (IRB: TMU-N201603037) underwent a CT scan in the supine position of free breathing using Brilliance CT Big Bore^TM^ (Philips, Amsterdam, The Netherlands). The total group of patients was divided into two groups: those with a customized PERSBRA and those without PERSBRA.

### 2.3. Delineation of Target Volume and Organs at Risk

Table 1 shows the characteristics of these 24 patients. The mean age of the patients was 51.7 years (range 35–82). The clinical target volume (CTV) was based on the ESTRO target guidelines [23]. The same radiation oncologist delineated all CT images of visible left breast tissue, which shrunk by 5 mm near the skin. The mean CTV of these 24 patients without and with PERSBRA was 385.0 ± 134.6 cc and 388.1 ± 145.3 cc, respectively. The planning target volume (PTV) was based on the uncertainty of the setup and the respiratory motion of the 5 mm margin of the CTV expansion and decreased by 5 mm near the skin. The mean PTV of these 24 patients without and with PERSBRA was 538.0 ± 158.1 cc and 520.2 ± 162.5 cc, respectively. The mean heart volume of these 24 patients without and with PERSBRA was 476.7 ± 64.1 cc and 471.7 ± 61.5 cc, respectively.

The autosegmentation function of the treatment plan was used to define the contours of the left lung. The radiation oncologist used the heart atlas published in 2011 [24] to delineate the contours of the heart and the substructure of the heart (LAD artery, LV, and right coronary artery). All contours are presented in Figure 2; of these, PTVs for (a) without PERSBRA and (b) with PERSBRA are shown in red. From Figure 2, it can be seen that the patient’s left breast was held in a prone position in the group with PERSBRA in the CT image.

### 2.4. Treatment Planning

This study plan generated 6 MV energy using the Pinnacle3 version 9.8C treatment planning system (Phillips Healthcare, Andover, MA, USA). In both the groups without and with PERSBRA in the same patients, the shape and size of PTV were used to administer 4–6 fields of IMRT; of these, the tangential field was fixed for two angles of the IMRT field, while the angles of 2 to 4 irradiation fields were 10 to 20 degrees above the tangential field and were other IMRT fields (Figure 3).

All treatment plans were based on clinical guidelines for whole-breast radiotherapy [25], and the prescribed dose was 5000 cGy in 25 fractions to the target of the whole breast (PTV). The evaluation of the treatment plan was based on the Radiation Therapy Oncology Group (RTOG) guidelines [26], in which 95% of the prescribed dose should cover at least 95% of the PTV, and the dose to the organs at risk should be as low as possible. In healthy tissue, the mean dose of the left lung was 18 Gy, the mean dose of the heart was 4 Gy, the mean dose of LAD was 20 Gy, and the mean dose of the right breast was 2 Gy [27].

### 2.5. Plan Evaluation Parameters

For PTV coverage, V_95%_ (relative volume that received <95% of the prescribed dose) was analyzed without and with PERSBRA treatment planning. The PTV homogeneity index (HI) was calculated as follows [28]:(D_2_ − D_98_)/Dp
where D_2_ and D_98_ were the minimum dose that covered 2% and 98% of the PTV volume, and Dp represented the prescribed dose. For the PTV conformity index (CI) = V_95%RI_/TV [29], V95%RI represented the volume covered by 95% of the prescribed dose, and TV the PTV volume.

In this study, we analyzed the mean dose of the heart, the maximum dose, and the mean dose of the Rt breast. For LAD, the maximum dose, V_30Gy_, V_40Gy_, and the mean dose were analyzed. For LV, V_5Gy_, V_23Gy_, and the mean dose were analyzed. The maximum dose and the mean dose were analyzed for the right coronary artery (RCA). The parameters V_5Gy_, V_10Gy_, V_20Gy_, and the mean lung dose were analyzed for the left lung. V_xGy_ represents the volume of organ that receives x Gy dose.

### 2.6. Statistical Analyses

The Wilcoxon signed-rank test was performed to compare differences between cases without and with PERSBRA treatment planning parameters. The analysis software was R version 4.0.3 (Vienna, Austria). The level of statistical significance was set as a *p*-value < 0.05 for all tests. The median is presented for the data in the study due to the small number of patients.

## 3. Results

In PTV coverage (Table 2), without or with PERSBRA, V_95%_ achieved 95% of the prescribed dose, which was 95.94 ± 1.12 and 95.76 ± 1.11 without and with PERSBRA, respectively. To evaluate the conformity of the plan, the conformity index (CI) was calculated. Lower CI was found among patients with PERSBRA compared to those without PERSBRA (1.28 ± 0.15 vs 1.34 ± 0.11). An HI value close to 0 represents a more homogeneous dose within the PTV; without PERSBRA, HI = 0.148 ± 0.03, and with PERSBRA, HI = 0.169 ± 0.05.

The use of PERSBRA combined with IMRT decreased the mean heart dose in 24 patients with left breast cancer from 382.3 cGy to 221.8 cGy. which was a 42% reduction. The maximum LAD dose decreased from 4903.9 cG to 3608.2 cGy, which was a 26.4% reduction; the mean LAD dose decreased from 1621.6 cGy to 859.0 cGy, which was a reduction of 47.0%. The mean dose of LV decreased from 604.3 cGy to 277.5 cGy, which was a reduction of 54.1%. The volume of the left lung that received 20 Gy (V_20_), 10 Gy (V_10_), 5 Gy (V_5_) decreased from 17.1% to 13.2%, 24.8% to 19.9%, 34.3% to 28.9%, respectively. The mean dose of the left lung decreased from 946 cGy to 725.2 cGy. Therefore, the PERSBRA tool can decrease the lung volume receiving 20 Gy, 10 Gy, and 5 Gy by 15.7% to 22.8% (Table 3).

The comparison of dose volume histograms (DVHs) with and without PERSBRA is summarized in Table 4 and Figure 4 and Figure 5. There were no differences in the volume of CTV (*p* = 0.992), PTV (*p* = 0.617), and heart (*p* = 0.721) volume between patients without and with PERSBRA (Table 4). The 24 patients presented a significantly lower (*p* < 0.001) mean heart dose, maximum LAD dose, mean LAD dose, LAD V_30Gy_, LAD V_40Gy_, mean LV dose, LV V_5Gy_, and LV V23Gy in the group with PERSBRA using the IMRT (Table 4). The V_20Gy_, V_10Gy_, V_5Gy_, and mean lung dose of the left lung were also significantly lower in the PERSBRA group (Table 4, Figure 5). There was no difference in the maximum dose of RCA and the mean dose between the patients without and with PERSBRA (Table 4, Figure 5).

## 4. Discussion

From the CT images of the patients in the two groups (Figure 2), using PERSBRA resulted in the CT image of the left breast appearing similar to that in the prone position, allowing organs at risk to be further away from the irradiation field. This finding is particularly true for the 2000 cGy isodose line, where it can be seen that 39% and 8% of the LAD volume received radiation in the cases without PERSBRA and with PERSBRA, respectively. At the 500-cGy isodose line, there was 22% and 7% of LV volume radiation in the groups without PERSBRA and with PERSBRA, respectively. From the DVH in Figure 4, it can be seen that PERSBRA can significantly decrease the dose for the LAD, LV, heart, and left lung.

A CI value close to 1 indicates a higher conformity dose within the PTV coverage. Regarding the coverage of PTV without and with PERSBRA in this study (Table 2), the CI with PERSBRA (CI = 1.28) was lower than that without PERSBRA (CI = 1.34) as the usage of PERSBRA causes the breast to be concentrated, thus increasing the conformity of PTV coverage. The PTV HI with PERSBRA was not better than that without PERSBRA due to the more stringent restrictions on organs at risk. However, the HI with PERSBRA was 0.169, which was within the range (0–0.5) recommended by Lingling Yan [30].

Table 3 shows that in all patients with left breast cancer who used PERSBRA, the dose in the organs at risk (lung and heart) was reduced by at least 15%. In particular, the mean heart dose decreased by 42%. Therefore, the probability of major cardiovascular disease can be decreased by 11.8% (a reduction from 382.3 cGy to 221.8 cGy) [5]. PERSBRA will provide a new radiotherapy option for patients undergoing left-body whole-breast radiotherapy who are not suitable for DIBH.

From Figure 4 and Table 4, it can be seen that only the mean dose of the right breast with PERSBRA was higher than that without PERSBRA, and the doses in other organs at risk were all lower than in cases without PERBRA. Although the mean dose of the right breast with PERSBRA was statistically higher than that without PERSBRA (*p* = 0.032), its median dose of 57.8 cGy was still within the evaluation constraint of the mean dose of the right breast of 200 cGy [27]. Stovall M et al. pointed out that patients < 40 years of age whose contralateral breast receives >100 cGy are likely to develop second primary breast cancer [31]. In PERSBRA with IMRT, the median contralateral breast dose < 100 cGy can decrease the probability of developing a second primary breast cancer.

Table 4 also analyzed RCA with and without PERSBRA and found no differences in maximum dose (*p* = 0.665) and mean dose (*p* = 0.452). However, the dose in the RCA must still be minimized to decrease the risk of coronary artery disease [5].

Most articles evaluated the dose effects of left whole-breast radiation therapy on organs at risk and mainly compared the mean dose of the heart and the mean dose of LAD between different treatments [13,32,33,34,35,36,37,38], and there were no strict dose restrictions for the mean dose of the heart and the LAD. In a 2019 article published by the German Society of Radiation Oncology, it was recommended that the limits of the heart dose [39] in breast cancer patients receiving radiotherapy should be as follows: mean whole heart dose 250 cGy, mean dose < 250 cGy, LV mean LV dose < 300 cGy, LV V_5Gy_ < 17%, LV V_23Gy_ < 5%, LAD dose < 1000 cGy, LAD V_30Gy_ < 2%, and LAD V_40Gy_ < 1%. Our study proved that PERSBRA combined with IMRT can achieve the dose constraints for the heart, LAD, and LV.

Although the incidence of radiation-induced pneumonitis due to radiotherapy for breast cancer is low (approximately 1–5%) [40], an increased lung dose will increase the probability of developing radiation-induced pneumonitis [41]. From Table 4, it can be seen that PERSBRA can significantly decrease the mean dose of the left lung and the volumes of V_20Gy_, V_10Gy_, and V_5Gy_. IMRT with PERSBRA can decrease the probability of developing radiation-induced pneumonitis.

In the study of Varga et al., it has been found that prone positioning is more beneficial for the reduction of heart and lung radiation exposure [42]. They found mean heart dose = 218 cGy and mean LAD dose = 1106 cGy with prone positioning. For our study, using PERSBRA with IMRT, the mean heart dose was 221 cGy and the mean LAD dose was 859 cGy. PERSBRA with IMRT can achieve results similar to those of prone positioning in reducing the heart exposure dose.

DIBH is a better method for reducing heart or lung irradiation in supine radiotherapy [13]. Lin et al. published a paper in 2019 on dose evaluation in DIBH [43]. They found that in 63 patients with left breast cancer treated with DIBH, the mean heart dose = 299.3 cGy, the mean left lung dose = 895.2 cGy, left lung V_20_ = 17.4%, left lung V_10_ = 24.5%, and left lung V_5_ = 33.2%. This study used PERSBRA and the free breathing method and a mean heart dose = 221.8 cGy, mean left lung dose = 725.2 cGy, left lung V_20_ = 13.2%, left lung V_10_ = 19.9%, left lung V_5_ = 28.9% (Table 3). PERSBRA with IMRT can achieve the same results as DIBH in reducing the dose of organs at risk.

There are some limitations of this study due to its retrospective nature. As PERSBRA is an innovative design, the sample size of this study is low. In the future, the use of PERSBRA should be investigated in more patients to prove the benefits of PERSBRA for organs at risk. However, the sample size of this study is statistically significant. According to the study of Chen et al. [22], the solid PERSBRA may increase the surface dose, but the risk could be reduced by modifying the PERSBRA design from solid mode to mesh mode. Furthermore, the same radiation oncologist delineated the contours of the target and organs at risk in this study. As there is subjective variation between different physicians, deep learning autosegmentation contours for all organs at risk can be used in future studies to obtain robust PERSBRA results [44].

## 5. Conclusions

Radiation exposure to the heart is a risk factor for radiation-induced heart disease. The results of the study showed that PERSBRA with IMRT greatly decreases the dose in organs at risk (lung, heart, LV, and LAD). This study also found that PERSBRA with IMRT can achieve similar results to deep inspiration breath-hold radiotherapy (DIBH) and prone positioning in terms of reducing the heart radiation dose. For patients who undergo whole-breast radiotherapy but cannot tolerate DIBH, the use of PERSBRA with IMRT may be an alternative option for heart-sparing radiotherapy.

## Figures and Tables

**Figure 1 jpm-12-01368-f001:**
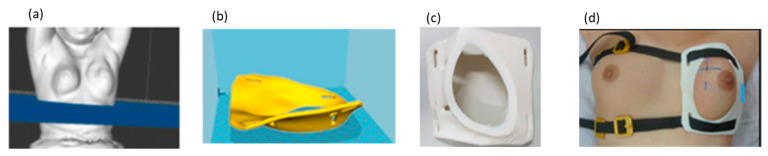
The patient treatment procedure with PERSBRA. (**a**) Acquisition of 3D breast shape by 3D scanner. (**b**) Transfer of the breast contour to the image system file. (**c**) 3D printing of PERSBRA. (**d**) Treatment with PERSBRA.

**Figure 2 jpm-12-01368-f002:**
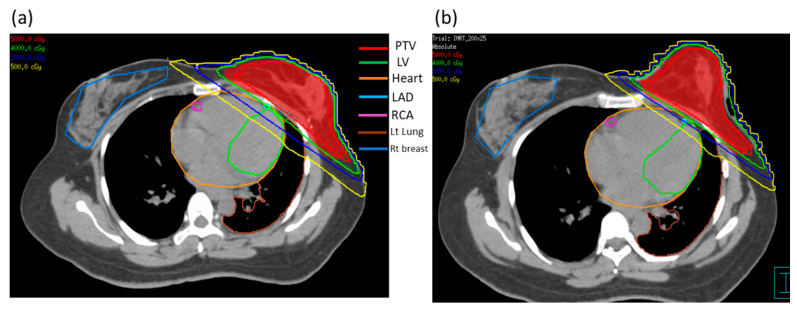
The dose distribution between cases without (**a**) and with PERSBRA (**b**). The red line is the isodose line of 5000 cGy; the blue line is 2000 cGy, and the yellow line is 500 cGy. Abbreviations: PTV = planning target volume, LV = left ventricle, LAD = left anterior descending artery, RCA = right coronary artery.

**Figure 3 jpm-12-01368-f003:**
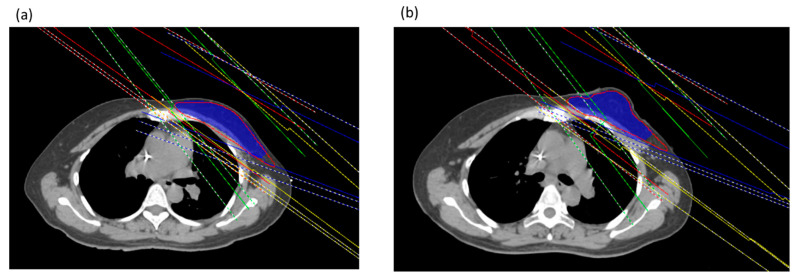
Beam arrangement between cases without (**a**) and with PERSBRA (**b**). The red line is the isodose line of 5000 cGy; the blue color is PTV. The beam arrangement was two tangential IMRT fields and two gantries tilted up 10–20° IMRT fields.

**Figure 4 jpm-12-01368-f004:**
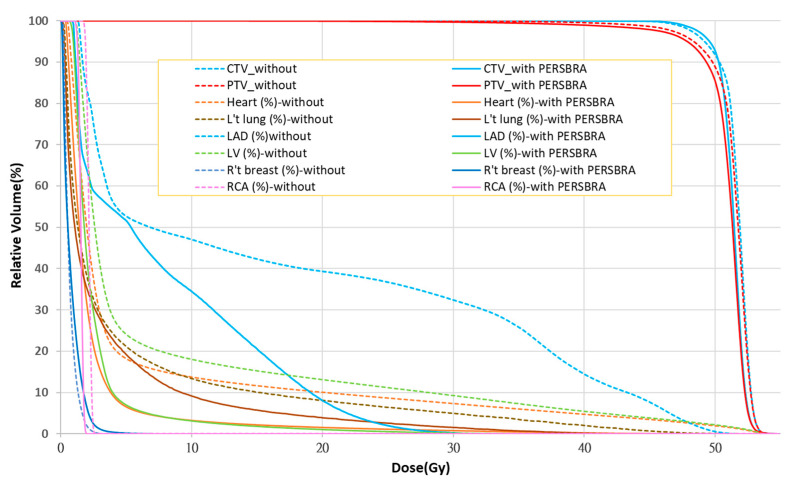
Comparison of the dose volume histograms (DVHs) between cases with and without PERSBRA. The solid line represents with PERSBRA; the dashed line represents without PERSBRA.

**Figure 5 jpm-12-01368-f005:**
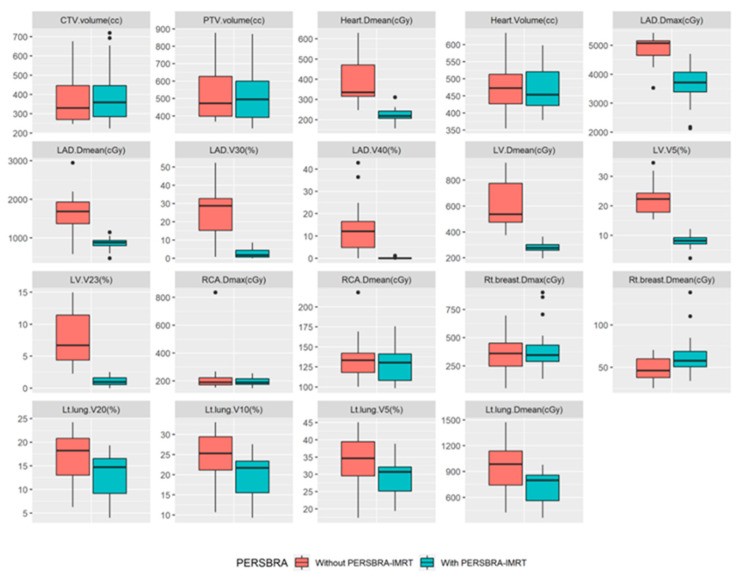
Box plots of target and organs at risk without PERSBRA versus with PERSBRA. Abbreviations: CTV = clinical target volume, PTV = planning target volume, LV = left ventricle, LAD = left anterior descending artery, RCA = right coronary artery.

**Table 1 jpm-12-01368-t001:** Patients’ characteristics.

	Without PERSBRA-IMRT	With PERSBRA-IMRT
	Mean ± SD (Range)	Mean ± SD (Range)
Age (in years)	51.7 ± 11.0 (35–82)	
CTV		
Volume (cc)	385.0 ± 134.6 (247.2–676.0)	388.1 ± 145.3 (223.7–719.3)
PTV		
Volume (cc)	538.0 ± 158.1 (367.4–816.2)	520.2 ± 162.5 (328.1–869.7)
Heart		
Volume (cc)	476.7 ± 64.1 (354.8–595.3)	471.7 ± 61.5 (379.8–598.3)

All data (*n* = 24) are presented as mean ± SD (range), Abbreviations: CTV = clinical target volume; PTV = planning target volume.

**Table 2 jpm-12-01368-t002:** Comparison of the planning target volume coverage without and with PERSBRA in intensity-modulated radiation therapy.

	Without PERSBRA-IMRT	With PERSBRA-IMRT
	Mean ± SD	Mean ± SD
PTV D_95(%)_	95.94 ± 1.12	95.76 ± 1.11
PTV CI	1.34 ± 0.11	1.28 ± 0.15
PTV HI	0.148 ± 0.03	0.169 ± 0.05

All data (*n* = 24) are presented as mean ± SD, PTV = planning target volume, Dx% = minimum dose received by x% of volume, CI = conformity index, HI = homogeneity index.

**Table 3 jpm-12-01368-t003:** Dosimetric comparison of organs at risk between patients with and without PERSBRA in IMRT (mean ± SD, *n* = 24).

Organ at Risk Mean ± sd	Without PERABRA-IMRT	With PERSBRA-IMRT	Reduction (%)
Heart Dmean (cGy)	382.3 ± 100.5	221.8 ± 33.9	42.0%
LAD			
Dmax (cGy)	4903.9 ± 419.4	3608.2 ± 665.2	26.4%
Dmean (cGy)	1621.6 ± 511.8	859.0 ± 154.8	47.0%
LV			
Dmean (cGy)	604.3 ± 163.1	277.5 ± 43.0	54.1%
Left lung			
V_20_ (%)	17.1 ± 5.3	13.2 ± 4.5	22.8%
V_10_ (%)	24.8 ± 6.2	19.9 ± 5.2	19.8%
V_5_ (%)	34.3 ± 7.3	28.9 ± 5.1	15.7%
Dmean (cGy)	946.1 ± 268.9	725.2 ± 181.4	23.3%

Note: Reduction (%) = {(with PERSBRA-without PERSBRA)/without PERSBRA} × 100%. Dmean = mean dose (cGy), Dmax = maximum dose (cGy), LAD = left anterior descending artery, LV = left ventricle, Vx = volume (%) receiving x dose (Gy).

**Table 4 jpm-12-01368-t004:** Dose volume metrics without and with PERSBRA treatment plans using the Wilcoxon signed-rank test.

Metric	Without PERSBRA-IMRT	With PERSBRA-IMRT	*p*-Value
	Median (IQR)	Median (IQR)	
CTV			
Volume (cc)	329.3 (176.6)	358.9 (162.2)	0.992
PTV			
Volume (cc)	471.8 (228.7)	495 (208.8)	0.617
Heart			
Dmean (cGy)	335.1 (155.5)	218.5 (35.6)	<0.001 *
Volume (cc)	472.7 (86.4)	453.6 (98.8)	0.721
LAD			
Dmax (cGy)	5068.6 (502.9)	3716.4 (679.1)	<0.001 *
Dmean (cGy)	1684.7 (556.4)	873.7 (137.8)	<0.001 *
V_30_ (%)	28.7 (17.6)	1.6 (3.8)	<0.001 *
V_40_ (%)	12.1 (11.7)	0 (0)	<0.001 *
LV			
Dmean (cGy)	536.1 (300.2)	277 (42.3)	<0.001 *
V_5_ (%)	22.3 (6.4)	8.1 (2.2)	<0.001 *
V_23_ (%)	6.7 (7)	0.9 (1)	<0.001 *
RCA			
Dmax (cGy)	190.6 (50.3)	189.4 (41.1)	0.665
Dmean (cGy)	133.2 (23.9)	130.2 (32.9)	0.452
Right breast			
Dmax (cGy)	358.5 (203.8)	345 (146.1)	0.798
Dmean (cGy)	46.0 (22.2)	57.8 (18.4)	0.032 *
Left lung			
V_20_ (%)	18.2 (7.7)	14.7(7.4)	0.006 *
V_10_ (%)	25.3 (8.2)	21.7(7.8)	0.004 *
V_5_ (%)	34.6 (9.9)	30.7 (7)	0.007 *
Dmean (cGy)	984.8 (392.9)	797.2 (296)	0.002 *

All data (*n* = 24) are presented as median (IQR), *: Wilcoxon signed-rank test *p*-value < 0.05. CTV = clinical target volume, PTV = planning target volume, LAD = left anterior descending artery, LV = left ventricle, RCA = right coronary artery, VxGy = volume of organ receiving x Gy dose.

## Data Availability

The data presented in this study are available on request from the corresponding author. The data are not publicly available due to patients’ privacy and medical ethics.

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
