# Peer review of "Retrospective Analysis for Dose Reduction to Organs at Risk with New Personalized Breast Holder (PERSBRA) in Left Breast IMRT"

_jpm, 2022, doi:10.3390/jpm12091368_

Round 1
Reviewer 1 Report
The paper is on using a fixation device "PERSBRA'' during left breast IMRT. the paper sounds scientific and can be published in this journal after several editing and corrections:
1. change ''left-sided 4 whole-breast radiation therapy" to "left breast IMRT" in the title
2. delete whole-breast radiation therapy and left anterior descending artery from the keywords and add organ at risk to the list
3. breast cancer can be diagnosed ''early'' (line 60 page 2) to ....''early stages''
4. if this study ''Chen et al.'' used PERSBRA, what is the novelty in your study? please explain the novelty of your study
5. line 98, page 2: consider these papers for reference:
10.32768/abc.202183192-202
10.1259/bjr.20180287
6. page 5 line 165: make the font color black
7. make fonts in the tables a bit smaller
8. page 10, line 321: did you make PERSBRA for each patient? then what about increasing the time and cost of patient fixation procedure as a limitation of this method?
9. conclusion is too short. explain more about the method used in this study
10. while the whole paper is about using PERSBRA, you concluded that IMRT combined...
Rewrite this sentence: PERSBRA with IMRT ...
Author Response
- change ''left-sided 4 whole-breast radiation therapy" to "left breast IMRT" in the title
Response: Thank you for the suggestion. We have revised the manuscript title.
Change in Manuscript:
Please see line 2 -3 : Retrospective analysis for dose reduction to organs at risk with new personalized breast holder (PERSBRA) in left breast IMRT
- delete whole-breast radiation therapy and left anterior descending artery from the keywords and add organ at risk to the list
Response: Thank you for the comment. We have deleted whole-breast radiation therapy and left anterior descending artery from the keywords and added organs at risk to the keywords list.
Change in Manuscript:
Please see line 51 -52 : Keywords: breast cancer, personalized breast holder, organs at risk, intensity-modulated radiation therapy
- breast cancer can be diagnosed ''early'' (line 60 page 2) to ....''early stages''
Response: Thank you for the comment. We have revised the associated text.
Change in Manuscript:
Please see line 58-59 : Breast conserving surgery combined with postoperative radiation therapy is the recommended treatment for patients with diagnosed to early stages .
- if this study ''Chen et al.'' used PERSBRA, what is the novelty in your study? please explain the novelty of your study
Response: According Chen et al. used PERSBRA in left breast cancers with conventional technique, our study retrospective used IMRT technique and compare more organs at risk for dose reduction, such as the right coronary artery, left ventricle, and right breast under personalized breast holder (PERSBRA) .
Change in Manuscript:
Please see line 2 -3 : Retrospective analysis for dose reduction to organs at risk with new personalized breast holder (PERSBRA) in left breast IMRT
- line 98, page 2: consider these papers for reference:
10.32768/abc.202183192-202
10.1259/bjr.20180287
Response: For reference “10.1259/bjr.20180287” the mean dose to heart are 2.9 ± 2.2 Gy, the mean dose to whole LAD are 17.8 ± 14 Gy, heart mean dose and LAD mean dose are our reference .
For reference” 10.32768/abc.202183192-202” are not found our reference list
- page 5 line 165: make the font color black
Response: Thank you for the suggestion. We have revised the manuscript line 165 font color.
Change in Manuscript:
Please see lines 165 : (Figure 3).
- make fonts in the tables a bit smaller
Response: Thank you for the comment. We have revised the tables in smaller fonts.
Change in Manuscript: Please see tables
Table 1: Patients characteristics
|
without PERSBRA-IMRT |
with PERSBRA-IMRT |
||
|
  |
Mean±SD (range) |
  |
Mean±SD (range) |
|
Age (in years) |
51.7 ±11.0 (35-82) |
||
|
CTV |
|
||
|
Volume (cc) |
385.0 ± 134.6 (247.2-676.0) |
388.1 ± 145.3 (223.7-719.3) |
|
|
PTV |
|||
|
Volume (cc) |
538.0 ± 158.1 (367.4-816.2) |
520.2 ± 162.5 (328.1-869.7) |
|
|
Heart |
|||
|
Volume (cc) |
476.7 ± 64.1 (354.8-595.3) |
471.7 ± 61.5 (379.8-598.3) |
|
Table 2. Comparison of the planning target volume coverage without and with PERSBRA in intensity-modulated radiation therapy
|
without PERSBRA-IMRT |
with PERSBRA-IMRT |
|
|
  |
Mean±SD |
Mean±SD |
|
PTV D95(%) |
95.94 ± 1.12 |
95.76 ± 1.11 |
|
PTV CI |
1.34 ± 0.11 |
1.28 ± 0.15 |
|
PTV HI |
0.148 ± 0.03 |
0.169 ± 0.05 |
Table 3. Dosimetric comparison of organs at risk between patients with and without PERSBRA in IMRT (mean ± SD, n = 24)
|
Organ at risk mean±sd |
Without PERABRA-IMRT |
With PERSBRA-IMRT |
Reduction(%) |
|
Heart Dmean (cGy) |
382.3 ± 100.5 |
221.8 ± 33.9 |
42.0% |
|
LAD |
|||
|
Dmax(cGy) |
4903.9 ± 419.4 |
3608.2 ± 665.2 |
26.4% |
|
Dmean(cGy) |
1621.6 ± 511.8 |
859.0 ± 154.8 |
47.0% |
|
LV |
|||
|
Dmean(cGy) |
604.3 ± 163.1 |
277.5 ± 43.0 |
54.1% |
|
Left lung |
|||
|
V20(%) |
17.1 ± 5.3 |
13.2 ± 4.5 |
22.8% |
|
V10(%) |
24.8 ± 6.2 |
19.9 ± 5.2 |
19.8% |
|
V5(%) |
34.3 ± 7.3 |
28.9 ± 5.1 |
15.7% |
|
Dmean(cGy) |
946.1 ± 268.9 |
725.2 ± 181.4 |
23.3% |
Table 4. Dose volume metrics without and with PERSBRA treatment plans using the Wilcoxon signed-rank test
|
Metric |
without PERSBRA-IMRT |
with PERSBRA-IMRT |
P value   |
|||
|
  |
median(IQR) |
median(IQR) |
||||
|
CTV |
||||||
|
Volume (cc) |
329.3 ( 176.6 ) |
358.9 ( 162.2 ) |
0.992 |
|||
|
PTV |
||||||
|
Volume (cc) |
471.8 ( 228.7 ) |
495 ( 208.8 ) |
0.617 |
|||
|
Heart |
||||||
|
Dmean (cGy) |
335.1 ( 155.5 ) |
218.5 ( 35.6 ) |
<0.001* |
|||
|
Volume (cc) |
472.7 ( 86.4 ) |
453.6 ( 98.8 ) |
0.721 |
|||
|
LAD |
||||||
|
Dmax (cGy) |
5068.6 ( 502.9 ) |
3716.4 ( 679.1 ) |
<0.001* |
|||
|
Dmean (cGy) |
1684.7 ( 556.4 ) |
873.7 ( 137.8 ) |
<0.001* |
|||
|
V30 (%) |
28.7 ( 17.6 ) |
1.6 ( 3.8 ) |
<0.001* |
|||
|
V40 (%) |
12.1 ( 11.7 ) |
0 ( 0 ) |
<0.001* |
|||
|
LV |
||||||
|
Dmean (cGy) |
536.1 ( 300.2 ) |
277 ( 42.3 ) |
<0.001* |
|||
|
V5 (%) |
22.3 ( 6.4 ) |
8.1 ( 2.2 ) |
<0.001* |
|||
|
V23 (%) |
6.7 ( 7 ) |
0.9 ( 1 ) |
<0.001* |
|||
|
RCA |
||||||
|
Dmax (cGy) |
190.6 ( 50.3 ) |
189.4 ( 41.1 ) |
0.665 |
|||
|
Dmean (cGy) |
133.2 ( 23.9 ) |
130.2 ( 32.9 ) |
0.452 |
|||
|
Right breast |
||||||
|
Dmax (cGy) |
358.5 ( 203.8 ) |
345 ( 146.1 ) |
0.798 |
|||
|
Dmean (cGy) |
46.0 ( 22.2 ) |
57.8 ( 18.4 ) |
0.032* |
|||
|
Left lung |
||||||
|
V20 (%) |
18.2 ( 7.7 ) |
14.7(7.4) |
0.006* |
|||
|
V10 (%) |
25.3 ( 8.2 ) |
21.7(7.8) |
0.004* |
|||
|
V5 (%) |
34.6 ( 9.9 ) |
30.7 ( 7 ) |
0.007* |
|||
|
Dmean (cGy) |
984.8 ( 392.9 ) |
797.2 ( 296 ) |
0.002* |
|||
- page 10, line 321: did you make PERSBRA for each patient? then what about increasing the time and cost of patient fixation procedure as a limitation of this method?
Response: Thank you for the comment. We used PESBRA for each patient in this study. It only took extra one minute to wear PERSBRA for each treatment. The cost of PERSBRA depends on 3D printer filament medical device grading.
- conclusion is too short. explain more about the method used in this study
Response: Thank you for the suggestion. We have revised the conclusion.
Change in Manuscript:
Please see line 336 - 342 : Radiation exposure to the heart is a risk factor of radiation-induced heart disease. The results of the study showed that PERSBRA with IMRT greatly decrease the dose in organs at risk (lung, heart, LV, and LAD). This study also found that PERSBRA with IMRT can achieve similar results as deep inspiration breath-hold radiotherapy (DIBH) and prone positioning in terms of reducing heart radiation dose. For patients undergoing whole-breast radiotherapy who cannot undergo DIBH, the use of PERSBRA with IMRT may be an alternative for heart sparing radiotherapy.
- while the whole paper is about using PERSBRA, you concluded that IMRT combined...
Rewrite this sentence: PERSBRA with IMRT ...
Response: Thank you for the suggestion. We have revised the conclusion sentence “PERSBRA with IMRT”.
Change in Manuscript:
Please see line 337 -342 : The results of the study showed that PERSBRA with IMRT greatly decrease the dose in organs at risk (lung, heart, LV, and LAD). This study also found that PERSBRA with IMRT can achieve similar results as deep inspiration breath-hold radiotherapy (DIBH) and prone positioning in terms of reducing heart radiation dose. For patients undergoing whole-breast radiotherapy who cannot undergo DIBH, the use of PERSBRA with IMRT may be an alternative for heart sparing radiotherapy.

Reviewer 2 Report
The study of Chen et al. compared the radiation exposition of the heart, coronary arteries and the ipsilateral lung during left whole breast irradiation (WBI) while using or not an individualized bra designed in the prone position for maintaining the prone position of the irradiated breast in the supine position. This technique sounds as a smart solution for utilizing the advantages while omitting the disadvantages of prone positioning for WBI. PERSBRA was fabricated on an individual basis utilizing a 3D printer. The dosimetry data were carefully and comprehensively analyzed. Nevertheless, the shortcomings of the manuscript need further input.
Major remarks
First, the reference (T.H. Chen, 2017) in line 93 is missing which could provide information on the below raised questions. Even if the manuscript included that reference, it should briefly refer to the following data: How was PERSBRA fabricated, what kind of material and technology was used, what were the radiation attenuation properties, how did it influence radiation treatment planning and delivery, was it biocompatible, etc.
How was the repositioning of PERSPBRA itself ensured, how was it documented
How was patient re-positioning accuracy investigated, and what was the result as compared in the two groups (if at all there were 2 groups??).
How was the radiogenic skin reaction between the 2 groups (if there were 2 groups??)
Did the protocol of RT planning differ between the 2 groups? The use of IMRT makes it risky to compare outcome between the 2 groups as compared to that if just conventional tangential fields were used.
How were the 2 groups of patients selected? Were these 2 cohorts without selection? Was the control group a matched population? Which patients qualified for having the PERSBRA device?
A most important conclusion of the study was that the distance between the heart and breast was increased by the use of the PERSBRA technique, however, no distances were measured. The other conclusions were not well-based either.
Minor remarks
In the Abstract the following sentence (line 36) „We compared the dose differences in target coverage and organs at risk with and without PERSBRA.” is misleading suggesting that these data were compared in the same patient (or was it??). The conclusions are inadequate since the authors did not study either the distance between the heart and breast nor DIBH or the consequences of radiogenic heart exposure.
Introduction: Lines 86-87: The statement „the criteria for DIBH are that the patient should be able to hold their breath for 30 seconds.” is not widely known hence a reference should be given.
Methods: The sections 2.1, 2.1, 2.4 are vaguely described while these would be the most interesting parts of the manuscript. It is not clear if there were 1 or 2 groups of patients etc.
Results: In line 206, the following sentence „The CI value close to 1 indicates a higher conformity dose within the PTV, without PERSBRA, CI = 1.34 ± 0.11, and with PERSBRA, CI = 1.28 ± 0.15.” is not correct since conformity is not a dose category, and the conslusion was the opposite than articulated in this sentence.
Discussion: This section repeats many of the results. Instead it, should put the findings into the context of individualized radiotherapy since there are so many approaches that might be more appropriate for some of the patients. It is not clear why the patients in this study with relatively small breast benefited from the prone-positioning of the PTV since that is more advantageous for patients with large breast; in fact in the study of Varga et al. prone positioning was de facto deteriorative regarding heart and LAD doses (Varga Z et al. Acta Oncol. 2014 Jan;53(1):58-64.) while in others prone positioning was more advantageous than the DIBH technique (Gaal S et al. Radiat Oncol. 2021 May 13;16(1):89.
Author Response
First, the reference (T.H. Chen, 2017) in line 93 is missing which could provide information on the below raised questions. Even if the manuscript included that reference, it should briefly refer to the following data: How was PERSBRA fabricated, what kind of material and technology was used, what were the radiation attenuation properties, how did it influence radiation treatment planning and delivery, was it biocompatible, etc.
Response: Thank you for the comment. We have revised our manuscript about PERSBRA information.
Change in Manuscript:
Please see line 88 - 89 : In a study by Chen, the PERSBRA material need more elasticity and biocompatible, they used thermoplastic elastomer (TPE) to fabricate a PERSBRA by 3D printer.
How was the repositioning of PERSBRA itself ensured, how was it documented
Response: Thank you for the comment. We marked the PERSBRA position on patient skin, then used cone beam computed tomography(CBCT) to check PERSBRA reproducibility.
How was patient re-positioning accuracy investigated, and what was the result as compared in the two groups (if at all there were 2 groups??).
Response: Thank you for the comment. In our study, there are only one group patient to compare without and with PERSBRA dosimetric analysis . We used cone beam computed tomography(CBCT) to check the patient position accuracy with PERSBRA every week.
How was the radiogenic skin reaction between the 2 groups (if there were 2 groups??)
Response: Thank you for the comment. In our study, there are only one group with PERSBRA. After the whole course of treatment, some of patient with PERSBRA has slight grade 1 erythema of skin reaction.
Reference:
- CP Chen, CY Lin, CC kuo et al. Skin Surface Dose for Whole Breast Radiotherapy Using Personalized Breast Holder: Comparison with Various Radiotherapy Techniques and Clinical Experiences. Cancers 2022, 14(13), 3205; https://doi.org/10.3390/cancers14133205
Did the protocol of RT planning differ between the 2 groups? The use of IMRT makes it risky to compare outcome between the 2 groups as compared to that if just conventional tangential fields were used.
Response: Thank you for the comment. The protocol between without and with PERSBRA planning are the same. According the conventional tangential fields, can reduce the heart mean dose, but can not reduce the LAD dose. In our study, we used IMRT to reduce heart and LAD dose.
How were the 2 groups of patients selected? Were these 2 cohorts without selection? Was the control group a matched population? Which patients qualified for having the PERSBRA device?
Response: Thank you for the comment. In our study, only one group of patients and two dosimetric analysis groups (without PERSBRA and with PERSBRA). All of patients joined the IRB study, and evaluated without and with PERSBRA planning results. If with PERSBRA had lower heart doses, then patients used PERSBRA for treatment.
A most important conclusion of the study was that the distance between the heart and breast was increased by the use of the PERSBRA technique, however, no distances were measured. The other conclusions were not well-based either.
Response: Thank you for the comment. We have revised our manuscript about conclusion.
Change in Manuscript:
Please see line 336-342 : Radiation exposure to the heart is a risk factor of radiation-induced heart disease. The results of the study showed that PERSBRA with IMRT greatly decrease the dose in organs at risk (lung, heart, LV, and LAD). This study also found that PERSBRA with IMRT can achieve similar results as deep inspiration breath-hold radiotherapy (DIBH) and prone positioning in terms of reducing heart radiation dose. For patients undergoing whole-breast radiotherapy who cannot undergo DIBH, the use of PERSBRA with IMRT may be an alternative for heart sparing radiotherapy.
In the Abstract the following sentence (line 36) „We compared the dose differences in target coverage and organs at risk with and without PERSBRA.” is misleading suggesting that these data were compared in the same patient (or was it??). The conclusions are inadequate since the authors did not study either the distance between the heart and breast nor DIBH or the consequences of radiogenic heart exposure.
Response: Thank you for the comment. In the abstract, we compared the dose differences in target coverage and organs at risk with and without PERSBRA in the same patient. We have revised the manuscript about conclusions.
Change in Manuscript:
Please see line 46-50: Using PERSBRA with IMRT greatly decreasing the dose to organs at risk (left lung, heart, left ventricle, and LAD). This study found that PERSBRA with IMRT can achieve similar results as deep inspiration breath-hold radiotherapy (DIBH) in terms of reducing heart radiation dose and the risk of developing heart disease in patients with left breast cancer who cannot undergo DIBH.
Introduction: Lines 86-87: The statement „the criteria for DIBH are that the patient should be able to hold their breath for 30 seconds.” is not widely known hence a reference should be given.
Response: Thank you for the comment. We have revised the associated text and added a reference.
Change in Manuscript:
Please see line 84 - 85 : the criteria for DIBH are that the patient should be able to hold their breath at least 20 seconds(Gaál, 2021)
Reference:
- Gaal, S., Kahan, Z., Paczona, V., Koszo, R., et al. (2021) Deep-inspirational breath-hold(DIBH) technique in left-sided breast cancer: various aspects of clinical utility. Radiat Oncol, 1.
Methods: The sections 2.1, 2.1, 2.4 are vaguely described while these would be the most interesting parts of the manuscript. It is not clear if there were 1 or 2 groups of patients etc.
Response: Thank you for the comment. We have revised the methods of the manuscript .
Change in Manuscript:
Please see line 109 -117 : This study used a personalized breast holder (PERSBRA) that fabricated with thermoplastic elastomer by 3D printer. The patient was first placed in a prone position and an infrared scanner was used to acquire the entire breast contours of the chest. This contour file was uploaded to the dedicated image processing system and a 3D printer was used to prepare a customized holder. The patient placed in a prone position while wearing the customized PERSBRA holder , undergoing daily treatment in the supine position (Figure 1). The patient was then placed in a supine position for every treatment, which showed better treatment reproducibility. PERSBRA was similar the prone position to far away the treatment field.
Please see line 128-129: The one group of patients were categorized into two groups: with a customized PERSBRA and without PERSBRA.
Please see line 161-165: In both the groups without and with PERSBRA in same patients, the shape and size of PTV were used to administer 4–6 fields of IMRT; of these, the tangential field was fixed for two angles of the IMRT field, while the angles of 2 to 4 irradiation fields were 10 to 20 degrees above the tangential field were other IMRT fields(Figure 3).
Results: In line 206, the following sentence „The CI value close to 1 indicates a higher conformity dose within the PTV, without PERSBRA, CI = 1.34 ± 0.11, and with PERSBRA, CI = 1.28 ± 0.15.” is not correct since conformity is not a dose category, and the conslusion was the opposite than articulated in this sentence.
Response: Thank you for the comment. We have revised the results of the manuscript.
Change in Manuscript:
Please see line 205 - 207 : To evaluate plan conformity, conformity index (CI) was calculated. Lower CI was found among patients with PERSBRA comparing to those without PERSBRA (1.28±0.15 vs 1.34± 0.11).
Discussion: This section repeats many of the results. Instead it, should put the findings into the context of individualized radiotherapy since there are so many approaches that might be more appropriate for some of the patients. It is not clear why the patients in this study with relatively small breast benefited from the prone-positioning of the PTV since that is more advantageous for patients with large breast; in fact in the study of Varga et al. prone positioning was de facto deteriorative regarding heart and LAD doses (Varga Z et al. Acta Oncol. 2014 Jan;53(1):58-64.) while in others prone positioning was more advantageous than the DIBH technique (Gaal S et al. Radiat Oncol. 2021 May 13;16(1):89.
Response: Thank you for the comment. We have revised the discussion of the manuscript.
Change in Manuscript:
Please see line 311 - 316 : In the study of Varga Z et al. has been found prone positioning is more benefit for the reduction of heart and lung radiation exposure(Varga, 2014). They found heart mean dose=218cGy and LAD mean dose=1106cGy in prone positioning. For our study used PERSBRA with IMRT, heart mean dose was 221cGy and LAD mean dose was 859cGy. PERSBRA with IMRT can achieve the similar results as prone positioning in reducing the heart exposure dose.
Round 2
Reviewer 2 Report
Thank you for revising the manuscript. Much has been done. The revised version is much clearer; the description of the methods is improved. A major shortcoming still exists: the physical and radiation characteristics of the material used for the fabrication of PERSBRA together with the specification of the 3D printer and more details about the procedure are needed. Actually, the reference which could help in part in this respect, and was provided in the response is not included (CP Chen, CY Lin, CC kuo et al. Skin Surface Dose for Whole Breast Radiotherapy Using Personalized Breast Holder: Comparison with Various Radiotherapy Techniques and Clinical Experiences. Cancers 2022, 14(13), 3205; https://doi.org/10.3390/cancers14133205). Which mash type was applied for PERSBRA? The mentioned paper raises the question on skin reaction. This is why it would be important to refer to radiodermatitis which is included in the response to the reviewer but not in the paper; the relevant questions by the reviewer need to be answered in the manuscript if accepted, in the present version these answers are not included. Please note that the order of references needs correction.
